# Communication-Traffic-Assisted Mining and Exploitation of Buffer Overflow Vulnerabilities in ADASs

**Yufeng Li [1,2], Mengxiao Liu [1], Chenhong Cao [1,2] and Jiangtao Li [1,2,*]**

[1] School of Computer Engineering and Science, Shanghai University, Shanghai 200444, China
[2] Purple Mountain Laboratories, Nanjing 211100, China
[*] Correspondence: lijiangtao@shu.edu.cn

**Abstract:** Advanced Driver Assistance Systems (ADASs) are crucial components of intelligent vehicles, equipped with a vast code base. To enhance the security of ADASs, it is essential to mine their vulnerabilities and corresponding exploitation methods. However, mining buffer overflow (BOF) vulnerabilities in ADASs can be challenging since their code and data are not publicly available. In this study, we observed that ADAS devices commonly utilize unencrypted protocols for module communication, providing us with an opportunity to locate input stream and buffer data operations more efficiently. Based on the above observation, we proposed a communication-traffic-assisted ADAS BOF vulnerability mining and exploitation method. Our method includes firmware extraction, a firmware and system analysis, the locating of risk points with communication traffic, validation, and exploitation. To demonstrate the effectiveness of our proposed method, we applied our method to several commercial ADAS devices and successfully mined BOF vulnerabilities. By exploiting these vulnerabilities, we executed the corresponding commands and mapped the attack to the physical world, showing the severity of these vulnerabilities.

**Keywords:** advanced driver assistance systems; buffer overflow vulnerability; communication traffic; intelligent vehicles





## 1. Introduction

Advanced Driver Assistance Systems (ADASs) equip vehicles with abundant capabilities, including blind spot monitoring, automatic emergency braking, pedestrian collision warning, and lane departure warning, thereby augmenting driving safety and improving driving experiences. The incorporation of numerous third-party machine learning libraries and other components into the code has made ADAS devices intricate software- and hardware-integrated systems, featuring voluminous code and varying implementation levels of diverse components. Consequently, the expanding functionalities of ADASs spawn a series of software vulnerabilities and pose challenges in terms of mining and exploitation.

Compared to infotainment systems and communication modules, such as Telematics-Box, vulnerabilities in ADASs pose a higher security risk. ADASs typically have external interfaces and are connected to a vehicle's power control system. Once a vulnerability is discovered, attackers can remotely hijack all vehicles equipped with the affected ADAS, resulting in disaster for both vehicle manufacturers and drivers. Additionally, once ADAS products are released, manufacturers often do not update the operating system, resulting in the kernel versions of most ADAS devices being outdated compared to the latest version. This leaves known vulnerabilities in the system open to direct exploitation, further elevating the security risk.

It has been shown that Advanced Driver Assistance Systems (ADASs) are vulnerable to various attacks that not only compromise their functionality but also pose significant risks to the physical world. For instance, an attacker could tamper with sensor inputs [1], such as radar or lidar, leading to false detections or blocking the system's ability to accurately

recognize obstacles [2]. By exploiting vulnerabilities in ADAS control algorithms, attackers can potentially manipulate steering, braking, or acceleration functions, endangering the lives of those in the vehicle and pedestrians [3]. Furthermore, software vulnerabilities enable adversaries to gain unauthorized access to the ADAS network and inject malicious commands or modify crucial system parameters, compromising the overall integrity and safety of the vehicle [4,5].

Buffer overflow (BOF) is a prevalent software vulnerability in ADASs that occurs when an application writes beyond its pre-allocated size during program execution. It can be used to launch denial-of-service attacks or to gain higher-order access privileges [6]. According to [7], BOF threats are the most growing and severe form of vulnerability in software today, and they have become an increasingly critical issue in network security. The code size of ADAS devices is massive, and ADAS devices use many third-party libraries whose quality varies. Therefore, BOF vulnerabilities are inevitable in ADASs.

In the realm of both traditional Internet and Internet of Things (IoT) devices, application programs commonly rely on open-source components and libraries. These components can be scrutinized through an examination of the corresponding open-source code to identify potential security risks such as BOF. Furthermore, open-source code can be evaluated using static code analysis tools, enabling automatic code scanning without the need for manual review. However, in the case of ADASs, manufacturers treat their environment and code as confidential commercial secrets, making it difficult to access data, code, and private communication protocols. Therefore, the BOF vulnerability mining of ADAS devices requires alternative approaches.

Analyzing BOF vulnerabilities in ADASs presents significant challenges. Firstly, obtaining the corresponding binary program from the device is difficult without publicly available documentation and source code. Decompiled code is only an approximation of pseudo-code, and most symbols, such as variables and function names, may be compressed and optimized. Furthermore, due to compiler optimization, some decompiled pseudo-code can be incomplete. Decompiled pseudo-code only reflects the program's behavior and is unsuitable for directly recompiling binary files, making it challenging to conduct BOF vulnerability mining using static code scanning tools.

Secondly, the code base for ADASs is huge, and, hence, a manual analysis is time-consuming. The scale of these code bases can encompass millions of lines of code, which makes it arduous for humans to fully comprehend the intricate interdependencies and linkages between different components of the code. Additionally, uncovering and evaluating potential vulnerabilities in code necessitate an in-depth understanding of the programming language, architecture, and intended system behavior. Moreover, a manual analysis requires an exhaustive inspection of every line of code, which is time-intensive, especially considering the several iterations required to ensure the complete coverage of the code base. Moreover, a manual analysis is susceptible to human error, and it can be time-consuming to identify subtle or concealed vulnerabilities that may escape human observation.

## 1.1. Related Work

It is known that BOF attacks have been causing serious security problems for decades; over 50 percent of today's widely exploited vulnerabilities are caused by BOF, and the ratio is increasing over time [8]. BOF-based attacks remain one of the most prevalent exploits to date, and as such, they are mostly an unsolved problem [9].

Due to the limited memory resources and generality of programming languages and operating systems, BOF poses unique risks to IoT devices. For example, a recent study evaluated the vulnerability to BOF attacks on operating systems in IoT, precisely executing the application on FreeRTOS and analyzing multiple attack methods [9]. This is because IoT devices typically do not choose substantial capacity memories to save power. However, the smaller the buffer, the easier it is to overflow. In addition, most programs used for IoT are written in C or C++, neither of which has a "garbage collection mechanism" to meet extra memory requirements, which also increases the risk of BOF vulnerabilities. At the

same time, these languages allow pointers, making it easier for hackers to determine the location of critical code in memory. For example, in 2018, researchers found a BOF vulnerability in the httpd component of Tenda routers, allowing attackers to conduct denial-of-service attacks by reverse-tracking the operation functions [10].

Moreover, IoT devices' operating systems and program dependency libraries are standard commercial products, and different manufacturers develop and maintain their codes. These code libraries may have vulnerabilities in different versions. Many embedded devices share the same operating system, application program libraries, and TCP/IP protocol stacks, and it could affect multiple devices if these shared components have vulnerabilities. For example, the chain of causes for Heartbleed is the input not being appropriately checked, which leads to too much data read—precisely, a massive number of bytes are read from the heap [11]. This affects many IoT devices using OpenSSL, which could lead to the leakage of sensitive information, such as private keys and user credentials.

As a particular type of IoT device, the components of intelligent connected vehicles share many similarities with traditional IoT devices, such as the reuse of customized systems and dependent libraries and most software being written in C/C++. These characteristics determine the high probability that vulnerabilities exist in these components. Many research results from various institutions also prove this point. In 2021, Tencent's Keen Security Lab conducted cybersecurity research on the intelligent connectivity system of Mercedes-Benz cars [4]. They discovered five vulnerabilities in the Head Unit and Hermes, including a BOF vulnerability. In 2017, Mickey Shkatov and two other researchers from McAfee announced at the Defcon 25 conference that they had found two separate BOF vulnerabilities in the remote information processing control units (TCUs) of Ford, BMW, Infiniti, and Nissan vehicles [12].

However, there remains a critical gap in the research on the mining and exploitation of BOF vulnerabilities in ADASs. Most academic research on ADASs focuses on the vulnerability of sensors and AI algorithms. For instance, Ben et al. showed that projecting non-depth objects in front of ADASs' cameras can cause Tesla Model X and Mobileye 630 systems to misperceive them as physical obstacles, resulting in unexpected actions [1]. However, Cao et al.'s LiDAR spoofing attack model put forward a potential safety threat to intelligent connected vehicles by tampering with the LiDAR system's input to the ADAS [2]. Thus, more research on vulnerability mining and exploitation for ADAS devices must be carried out.

ADAS devices have richer storage and processing capabilities than traditional IoT devices, using closed-source and proprietary protocols for communication. These situations are more likely to lead to BOF vulnerabilities. Most private protocols are unencrypted, allowing for the extraction of specific fields/features from traffic, which can help to quickly identify BOF vulnerabilities.

*1.2. Contribution*

The complex software environments of ADASs and their components, along with their large code base and extensive reliance on third-party components, make the process of mining and exploitation of BOF vulnerabilities challenging. In this work, we observed that ADAS devices commonly utilize unencrypted protocols for module communication. This provided an opportunity for us to extract information and features from raw communication traffic, enabling us to locate input stream and buffer data operations more efficiently. We achieved this by filtering out known protocols based on the captured device traffic, allowing us to more accurately locate data structures, input data passing, and buffer operations during the decompilation process. This approach reduces the workload of manual code auditing in the mining of BOF vulnerabilities. Based on the above situations, we proposed a communication-traffic-assisted ADAS BOF vulnerability mining and exploitation method.

In detail, our contributions are as follows:

1. We propose a method for mining and exploiting vulnerabilities in ADASs by analyzing their communication traffic. Our approach involves capturing and analyzing communication traffic, extracting feature fields, decompiling business applications, and locating data structures using these feature fields. By checking cross-references of the data structures, we can restore the entire data passing chain and analyze the data passing operations and buffer operations within this chain. Using these analyses, we can identify BOF vulnerabilities in ADASs.

2. To demonstrate the effectiveness of the proposed method and the severity of these vulnerabilities, we employed our method on several ADAS devices and mined the BOF vulnerabilities. Furthermore, we conducted a series of operations on a vehicle and captured the corresponding CAN messages. Subsequently, we replayed the communication traffic via the ADAS device and observed whether the vehicle repeated these operations. By exploiting the BOF vulnerabilities, we could execute console commands to replay CAN messages and map the attack to the physical world. Our experiment proves that BOF vulnerabilities in ADASs could interfere with the regular operation of intelligent connected vehicles.

*1.3. Organization*

This paper is organized as follows: First, we present basic knowledge on buffer overflow, the composition of ADAS software, and communication traffic characteristics (Section 2). Next, we propose a communication-traffic-assisted method for BOF vulnerability and exploitation (Section 3). Then, we conduct experiments on commercial ADAS products and actual vehicles to verify the method's feasibility (Section 4). Finally, we summarize the entire article (Section 5).

## 2. Preliminaries

*2.1. BOF Vulnerability*

BOF occurs when a computer fills a buffer with more data than its capacity, causing the overflowed data to overwrite legitimate data. The root cause of BOF attacks is an inherent flaw in modern computers that do not explicitly distinguish between data and code and can only rely on forward-compatible patches to mitigate the resulting damage [13].

There are two types of BOFs: heap overflow and stack overflow [14]. Both result from programs writing data to the heap/stack without correctly controlling the data size, but they differ in data structure and their role in program execution. Due to the stack's particular structure and role, stack overflow can more directly disrupt or control the program flow. In contrast, the heap structure is often more closely related to the system and version, making it more difficult to exploit.

Unmitigated BOFs commonly result in segmentation faults, causing the operating system to terminate the corresponding program, leading to program crashes and denial of service. By combining with return-oriented programming (ROP), constructing jump instructions, and gaining control of the operating system, arbitrary code/command execution can be achieved. In 1988, Robert Morris's Morris worm virus exploited a BOF vulnerability, causing more than 6000 network servers worldwide to crash [15].

As a specialized type of embedded system, it is hard for an ADAS to avoid BOF vulnerabilities. Once these vulnerabilities are detected, the reliability and security of ADAS devices will inevitably be compromised.

*2.2. Software Environment of ADASs*

By analyzing the software environment of ADASs, we constructed a more targeted BOF exploitation. For example, the position of the system function in libc is involved in the ROP construction of the RCE process. The position of the system function varies in different versions of libc. The software components of ADASs mainly include operating systems, runtime environments and dependent libraries, and artificial intelligence algorithms. In our research on market devices, we found that the operating systems of most devices usually

run on manufacturer-customized Linux systems or QNX systems, with Linux versions mostly between 2.6 and 4.13 and QNX between 6.5 and 7.1. Table 1 presents the operating systems and kernel versions of five commercial models of ADASs.

**Table 1.** Version information of the operating system and libc.

| Devices | Operating System and Kernel Version | Libc Version |
| --- | --- | --- |
| ADAS-1 | QNX Neutrino 6.6.0 SP1 | ldqnx-arm.le.so.2.1.0 |
| ADAS-2 | Linux cid 4.1.27-PLK #1 SMP PREEMPT x86_64 GNU/Linux | libc-2.22 |
| ADAS-3 | Linux (none) 4.14.0-xilinx #52 SMP PREEMPT armv7l GNU/Linux | libc-2.18 |
| ADAS-4 | Linux (none) 4.9.0-xilinx-svn403 #1 SMP PREEMPT armv7l GNU/Linux | libc-2.26 |
| ADAS-5 | Linux Ambarella 4.14.139 #1 SMP PREEMPT aarch64 Flexible Linux | libc-2.28 |

Software operating environments and dependencies mainly include two types. One is the most fundamental operating environment that the operating system provides, such as libc. The libc information is our main focus in security research and plays a vital role in exploiting buffer overflow vulnerabilities. Table 1 lists the libc version information of five commercial ADAS devices. Another type is the third-party component Software Development Kits (SDKs) that the application depends on. This type often varies among different manufacturers and lacks a unified quantifiable indicator. For example, SAIC's immotors ADAS uses the NVIDIA Xavier development environment, while Tesla's Autopilot has entirely independently developed related suites and does not use such third-party components.

Artificial intelligence algorithms are diverse, but when categorized, they mainly include supervised algorithms such as pattern recognition, SVM, regression algorithms, decision matrix algorithms, Ada-Boost, and unsupervised algorithms such as clustering [16].

### 2.3. Communication in ADASs

Through the study of over ten different commercial ADAS devices, we found that, unlike traditional IoT devices, open-source components are less commonly used in ADASs. Most ADAS programs choose to use commercial SDKs or to develop them entirely independently. During port scanning or analyses of the devices, we found that, apart from a few debugging ports (such as SSH and telnet), most open ports are related to the business and are listened to by the related business programs. Figures 1 and 2 show two commercial ADAS devices' opening ports and corresponding listening programs.

```
root@ice-unknown:~# ss -ltpu|grep LISTEN
tcp   LISTEN   0   0      *:4110              *:*            users:(("QtCarEVLogServi",pid=1980,fd=15))
tcp   LISTEN   0   0      *:4400              *:*            users:(("QtCarTMServer",pid=1997,fd=12))
tcp   LISTEN   0   0      *:4082              *:*            users:(("QtCarMediaServe",pid=2003,fd=12))
tcp   LISTEN   0   0      *:4500              *:*            users:(("NuanceServer",pid=2001,fd=16))
tcp   LISTEN   0   0      *:20564             *:*            users:(("ice-updater",pid=1961,fd=8))
tcp   LISTEN   0   0      127.0.0.1:domain         *:*           users:(("connmand",pid=1953,fd=13))
tcp   LISTEN   0   0      *:ssh               *:*            users:(("sshd",pid=1988,fd=3))
tcp   LISTEN   0   0      127.0.0.1:4567           *:*           users:(("hermes_vehicle_",pid=1956,fd=3))
tcp   LISTEN   0   0      *:4220              *:*            users:(("QtCar",pid=2000,fd=17))
tcp   LISTEN   0   0      *:4060              *:*            users:(("QtCarConnMan",pid=1979,fd=13))
tcp   LISTEN   0   0      *:4030              *:*            users:(("QtCarVehicle",pid=1989,fd=17))
tcp   LISTEN   0   0      *:4032              *:*            users:(("QtCarServer",pid=1974,fd=10))
tcp   LISTEN   0   0      *:4160              *:*            users:(("QtCarGpsManager",pid=1992,fd=15))
tcp   LISTEN   0   0      *:8002              *:*            users:(("valhalla_route_",pid=2008,fd=12))
tcp   LISTEN   0   0      *:4035              *:*            users:(("QtCarServer",pid=1974,fd=18))
tcp   LISTEN   0   0      *:25956             *:*            users:(("ice-updater",pid=1961,fd=7))
tcp   LISTEN   0   0      *:4070              *:*            users:(("QtCar",pid=2000,fd=19))
tcp   LISTEN   0   0      *:7654              *:*            users:(("QtCarServer",pid=1974,fd=14))
tcp   LISTEN   0   0      *:4170              *:*            users:(("QtCarVehicle",pid=1989,fd=12))
tcp   LISTEN   0   0      ::1:domain             :::*           users:(("connmand",pid=1953,fd=14))
tcp   LISTEN   0   0      :::ssh               :::*           users:(("sshd",pid=1988,fd=4))
root@ice-unknown:~#
```

**Figure 1.** Commercial ADAS devices' port opening and corresponding program listening-1.

```
root@m      s # netstat -anltpu | grep LISTEN
tcp    0    0 0.0.0.0:16801        0.0.0.0:*              LISTEN    676/mv_app
tcp    0    0 0.0.0.0:16802        0.0.0.0:*              LISTEN    651/mv_update
tcp    0    0 0.0.0.0:16803        0.0.0.0:*              LISTEN    676/mv_app
tcp    0    0 0.0.0.0:16804        0.0.0.0:*              LISTEN    676/mv_app
tcp    0    0 0.0.0.0:21           0.0.0.0:*              LISTEN    633/tcpsvd
tcp    0    0 :::23                :::*                   LISTEN    635/telnetd
root@m      s # |
```

**Figure 2.** Commercial ADAS devices' port opening and corresponding program listening-2.

We captured working communication traffic from multiple commercial ADAS devices and analyzed the traffic. We found that most communication traffic used the raw TCP protocol without encryption rather than other mature application layer protocols (such as HTTP and WebSocket). Figure 3 shows the communication traffic of a commercial ADAS device, and the situation of other commercial devices is similar to that of this device. Based on this, using traffic fields/characteristics to assist the binary static analysis is feasible. We improved the efficiency of BOF vulnerability mining with the assistance of communication traffic.

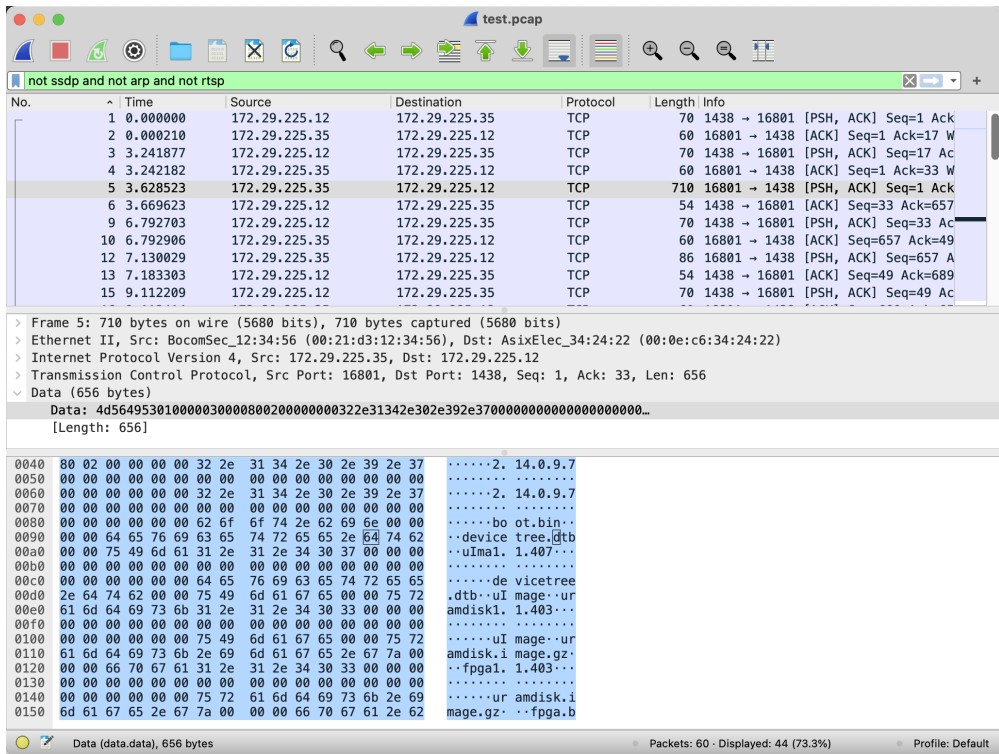

**Figure 3.** The communication traffic of commercial ADAS.

## 3. Traffic-Assisted BOF Vulnerability Mining and Exploitation Method in ADASs

This article proposes a traffic-assisted BOF vulnerability mining and exploitation method in Figure 4 to tackle the complexities of BOF vulnerability mining in ADAS devices. The first step is firmware extraction. Firmware extraction uses methods such as console commands or the reading of flash chips, and this is discussed in Section 3.1. The next step is a firmware and system analysis. We capture the communication traffic to obtain fields/features, collect system running information (such as listening status and background processes), and locate the ADAS manufacturer's application. This step is discussed in Section 3.2. Next, we identify the BOF risk points and mine vulnerabilities using available information, such as traffic. This step is discussed in Section 3.3. For detected BOF vulnerabilities, we validate their effectiveness, which is discussed in Section 3.4. Finally, Section 3.5 discusses the exploitation of BOF vulnerabilities, such as obtaining console access.

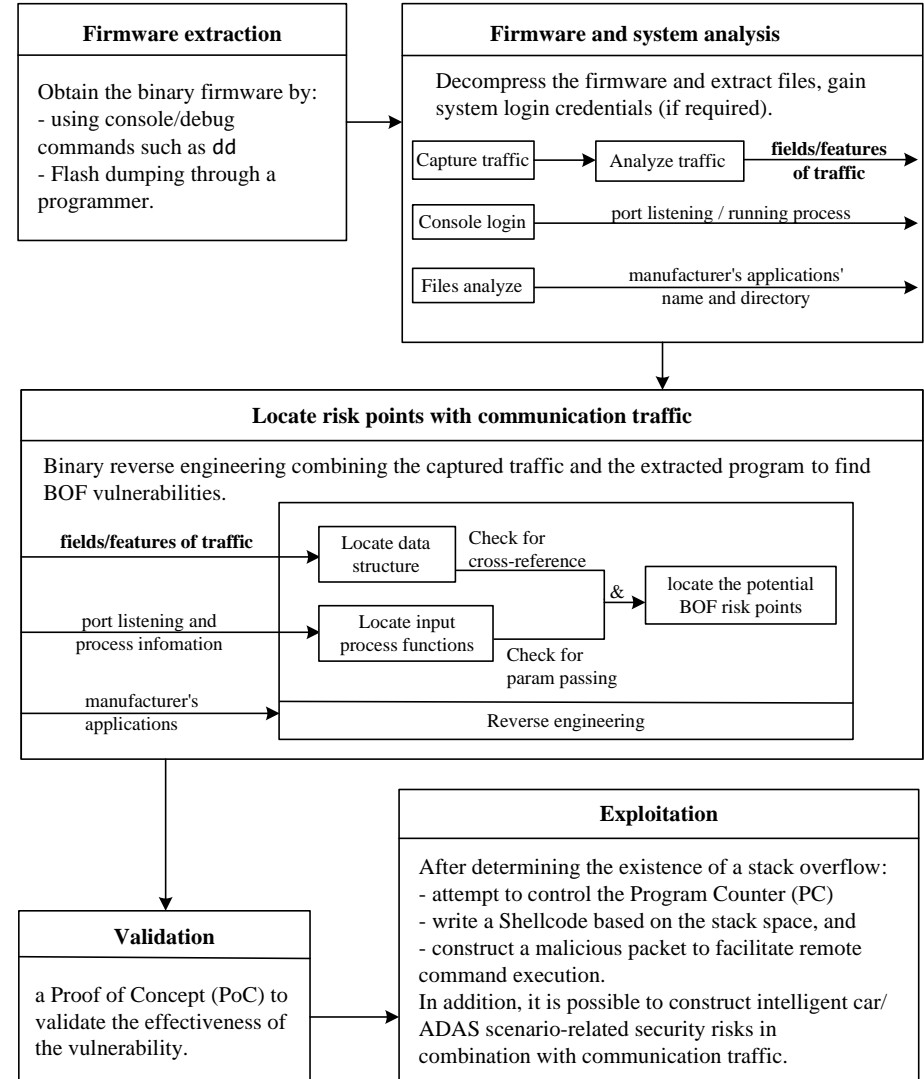

**Figure 4.** Traffic-assisted BOF vulnerability mining process.

### 3.1. Firmware Extraction

Firmware is a type of software that is written into hardware devices to control their application and system functions. We can obtain the corresponding binary files for the business program by extracting and analyzing the firmware. Thus, we proceed with the subsequent BOF vulnerability discovery.

There is no universal method for extracting firmware from devices. We list four practical firmware extraction methods from ADAS devices: update package interception, console command extracting, debug interface extracting, and flash dumping.

(1) Update package interception. We can capture the device-to-cloud communication traffic through the automatic update function of the corresponding PC software for the device. If the update package that we obtain is complete, we extract the complete firmware from the update package by combining the reverse engineering results of the client program.

(2) Console command extracting. We use a dictionary to brute-force Telnet/SSH login passwords for devices that can be accessed through the console. If we crack the password, we can log in to the console and use console commands to extract the corresponding binary file (such as dd) and transfer it back to the local machine (such as scp).

(3) Debug interface extracting. If a debug interface (such as SWD/JTAG) exists on the ADAS device, we use the testing interface to extract the firmware.

(4) Flash dumping. However, if none of the above methods can be used, we only remove the flash storage chip from the device board, connect the programmer to read the chip content, and write a program according to the file system to restore the binary file.

### 3.2. Firmware and System Analysis

In this section, we first decompress the extracted firmware and extract its files. If there is still no console login permission, we brute-force the password hash in `/etc/passwd`. Next, we proceed with three parallel tasks:

(1) Capturing and analyzing device communication traffic. In this step, we disguise our computer as a device node and connect it to the in-vehicle Ethernet to capture the communication traffic. Figure 3 shows part of the results. We write programs to analyze the captured traffic, extract potentially useful fields or features, and prepare for subsequent risk point localization.

(2) Logging in to the console and obtaining port listening and system process information. If we obtain console login permission, we should log in to the device and view the background processes in the system, especially the processes of third-party applications. This can help us to quickly locate the ADAS manufacturer's developed applications in the firmware.

(3) File analysis. If we still fail to obtain console permission, we must traverse the decompressed firmware folder and manually search for the corresponding ADAS manufacturer's program. Generally, applications developed by ADAS manufacturers appear in the form of libraries, modules, or executable files.

### 3.3. Locate Risk Points with Communication Traffic

BOF does not occur in all functions but typically happens in those that involve input/output interactions or data passing. In this subsection, we use reverse engineering with communication traffic to locate potential BOF vulnerabilities or risk points.

In the previous subsection, we acquired some fields/features of the communication traffic, port listening and process information, and corresponding ADAS manufacturer applications.

In this subsection, we decompile the ADAS manufacturer program, use the fields/features to search for binary program data fields, and attempt to locate data structures within the program. In addition, we determine the receiving and passing processes of data by searching for library functions, such as `socket`, `listen`, and `recv`, based on the port listening information and system-background process names.

As the definition of BOF indicates, failure to truncate user input appropriately is a necessary condition for the vulnerability. Therefore, risk point identification focuses on two aspects: (1) external input/output interaction and (2) whether the length is appropriately controlled during parameter copying and passing. By locating data structures through characteristic fields and checking for cross-references in the program while paying attention to data copying or passing processes, such as `gets`, `memcpy`, and `strcpy`, we efficiently detect BOF risk points or vulnerabilities.

### 3.4. Validation of BOF Vulnerability

If a BOF vulnerability is discovered, attempting to construct a POC program is possible. When validating a BOF vulnerability, the most vital aspect is constructing an appropriate payload for testing purposes. The payload is a block of data that contain adequate bytes to attempt to overwrite or modify the target application's buffer. Consideration is necessary when building the payload, particularly regarding the number of bytes to be used: it needs to be large enough to cover the buffer's end while small enough to fit legally in the buffer. A too-small payload will be inadequate to control the program's behavior, and if the payload is too large, it may prevent the program from crashing.

To verify the existence of a BOF vulnerability, we need to create one or a series of data packets. These packets correspond to the buffer's length or size based on the length requested in the program's pseudo-code and the actual buffer's size. We then use tools or write a program to send these packets to the target application program from our computer. The target application program typically has a BOF vulnerability if it crashes after receiving the payload. If the payload fails to cause the application to crash, we must adjust it and send it again until the vulnerability is discovered.

*3.5. Exploitation of BOF*

The validation of BOF vulnerabilities often only requires the construction of a data packet of the corresponding size or length. However, this is not enough to facilitate advanced exploitation and intrude into an ADAS more deeply. Generally, it is necessary to combine IDA Pro or other decompiling tools to determine the buffer size and its relative position. At the same time, one must use tools or manually write a payload to write executable code into the program's buffer. Additionally, the payload should contain special jump instructions to overwrite the control flow pointer and redirect program execution to the corresponding code for execution.

For example, typical BOF attack payloads include No Operation (NOP) slide and return-oriented programming attacks. With a NOP slide attack, an attacker adds a chain of NOP instructions before the malicious code, enabling them to slide to a specific location in the buffer and then jump to execute the malicious code. Another technique is ROP, where an attacker overwrites a return address to direct the program flow to a buffer section containing "gadgets"—instruction sequences (e.g., shellcode) that the attacker assembles to achieve arbitrary code execution.

## 4. Practical Results and Physical World Threats

This section applied the communication-traffic-assisted BOF vulnerability mining and exploitation method to over ten commercial ADAS devices. As a result, we successfully mined BOF vulnerabilities in two of them. Taking one of the commercial devices connected to a minibus as a case study, we elaborated on the details from firmware extraction to vulnerability exploitation based on the communication-traffic-assisted BOF vulnerability mining and exploitation method. Moreover, we designed an advanced exploit leveraging the communication traffic of the CAN protocol, demonstrating the disruptions in regular operations of the vehicle by exploiting the BOF vulnerability in ADAS devices and thereby confirming the effectiveness of our method.

*4.1. Traffic-Assisted BOF Vulnerability Mining in ADASs*

The first step is firmware extraction. We dismantled the binary data from the device's flash storage chip and used a programmer to read the chip. Then, we used binwalk to extract and decompress the firmware, which allowed us to obtain the files in the firmware.

Afterward, we conducted a firmware and system analysis. We conducted a port scan of the device and discovered that the device's SSH port (TCP 22) was open and using a username and password for login. We obtained the device's login credentials using hashcat to brute-force the `/etc/shadow` file extracted from the firmware. After logging into the device, we examined the system's port listening and corresponding processes, thus locating the binary program. By confirming the source through various means, such as the program name and digital signature, and combining it with the device's network communication flow, we determined that the "adas_app" program within the system directory was the relevant business program.

The next crucial step was to locate the risk points with the communication traffic. In this case, the "adas_app" program listens on two ports, 8080 and 8081. Combining the ADAS communication flow fields, we identified the corresponding functions for each port. Port 8081 was the data receive/send port, while port 8080 was the command receive/send port. We recorded the corresponding port listening functions and examined the

parameter passing process to analyze the subsequent data flow and more specific business processing functions.

Through a static analysis of the decompiled code of the monitoring section, we discovered that the 8081 port was mainly responsible for receiving the data portion corresponding to the command. We found no security vulnerabilities in this section. The 8080 port was mainly responsible for processing incoming commands. The data structure corresponding to the communication data packet of the 8080 port is shown below. The functions in this section mainly perform validity checks on the "check" field and enter different processing branches based on the "cmd" field.

```c
#pragma pack(1)
typedef struct {
    BYTE cmd;         // CMD No
    DWORD param;      // exec param
    DWORD version;    // version id: 1.2.1/1.2.3
    BYTE check;       // 0xaa
}ADAS_CMD;
#pragma pack()
```

Specifically, in this business program (in Figure 5), when the "cmd" is 6, the program will perform the following data processing operations:

**Figure 5.** Business program decompiled code (stack overflow pseudo-code).

(1) Receive 12 bytes, take the first byte to represent the "type", and multiply the following DWORD values, denoted as A, which represents the number of bytes to be received next.
(2) Enter the corresponding function for data reception, and in this function, first, receive A bytes, and then receive 4 bytes again as the data size, denoted as B.
(3) Receive B bytes of data.

During the execution of step (3), the program will allocate a receiving buffer in the stack, and the length of the buffer is less than 2M. During this process, if a data packet is artificially constructed and the value of the "B" field is set to be greater than 2M, the server will wait for and receive the length of data we set, leading to a stack BOF.

It is not difficult to see that the program will take the values of A and B from the network communication packet as the size of the buffer allocated subsequently. This defect can be exploited through malicious packet construction to cause a BOF. We attempt to construct a malicious packet of $0 \times 1,000,000$ bytes as a PoC program to trigger the corresponding vulnerability and observe the program's execution results. If a BOF vulnerability exists, the business program will crash and exit, and the corresponding port listener will also be released.

Before we executed the PoC program, we scanned the operating system (OS) opening ports and observed the port listening status of the program, as shown in Figure 6.

We executed the PoC program and scanned the open ports of the OS again (red border marked in Figure 7). The program's corresponding ports were closed, and the processes corresponding to the ports had also exited, indicating that a denial-of-service attack was successfully implemented.

```
nmap -T4 -A -v 192.168.198.198

Starting Nmap 7.92 ( https://nmap.org ) at 2021-09-02 12:37 ?D1ú±ê×?ê±??
NSE: Loaded 155 scripts for scanning.
NSE: Script Pre-scanning.
Initiating NSE at 12:37
Completed NSE at 12:37, 0.00s elapsed
Initiating NSE at 12:37
Completed NSE at 12:37, 0.00s elapsed
Initiating NSE at 12:37
Completed NSE at 12:37, 0.00s elapsed
Initiating ARP Ping Scan at 12:37
Scanning 192.168.198.198 [1 port]
Completed ARP Ping Scan at 12:37, 0.22s elapsed (1 total hosts)
Initiating Parallel DNS resolution of 1 host. at 12:37
Completed Parallel DNS resolution of 1 host. at 12:37, 0.11s elapsed
Initiating SYN Stealth Scan at 12:37
Scanning 192.168.198.198 [1000 ports]
Discovered open port 21/tcp on 192.168.198.198
Discovered open port 8080/tcp on 192.168.198.198
Discovered open port 22/tcp on 192.168.198.198
Discovered open port 8082/tcp on 192.168.198.198
Discovered open port 8081/tcp on 192.168.198.198
Completed SYN Stealth Scan at 12:38, 14.92s elapsed (1000 total ports)
Initiating Service scan at 12:38
```

**Figure 6.** Port open status before stack overflow.

```
nmap -T4 -A -v 192.168.198.198

Starting Nmap 7.92 ( https://nmap.org ) at 2021-09-02 12:39 ?D1ú±ê×?ê±??
NSE: Loaded 155 scripts for scanning.
NSE: Script Pre-scanning.
Initiating NSE at 12:39
Completed NSE at 12:39, 0.00s elapsed
Initiating NSE at 12:39
Completed NSE at 12:39, 0.00s elapsed
Initiating NSE at 12:39
Completed NSE at 12:39, 0.00s elapsed
Initiating ARP Ping Scan at 12:39
Scanning 192.168.198.198 [1 port]
Completed ARP Ping Scan at 12:39, 0.14s elapsed (1 total hosts)
Initiating Parallel DNS resolution of 1 host. at 12:39
Completed Parallel DNS resolution of 1 host. at 12:39, 0.00s elapsed
Initiating SYN Stealth Scan at 12:39
Scanning 192.168.198.198 [1000 ports]
Discovered open port 22/tcp on 192.168.198.198
Discovered open port 21/tcp on 192.168.198.198
Discovered open port 8082/tcp on 192.168.198.198
```

**Figure 7.** Port open status after stack overflow.

*4.2. Return-Oriented Programming and Remote Command Execution*

Return-oriented programming is a technique by which an attacker can induce arbitrary behavior in a program whose control flow they have diverted, without injecting any code [17]. Its core idea is to control the stack call by exploiting stack BOF, thereby hijacking the program control flow to execute specific machine language instructions, known as "gadgets". These gadgets can modify the return address of a function and redirect it to any desired location, allowing attackers to execute unauthorized instructions and gain system privileges for arbitrary code/command execution and various illegal operations.

In the previous subsection, we constructed a specific PoC to cause a denial-of-service attack on the ADAS device's business program. However, there is a better form of attack than a denial-of-service attack. In this subsection, we attempt to write a specific shellcode into a newly allocated buffer and use ROP techniques to control the EIP pointer, thereby executing the shellcode and gaining console access to the device.

By analyzing the assembly code, we determined that the address where the program writes data (see Figure 8) is 0x1490C bytes away from the return address on the stack (see Figure 9), with enough length to accommodate a shellcode.

**Figure 8.** Address where the program writes data.

**Figure 9.** Return address on the stack.

Next, we queried the protection mechanisms of this program (see Figure 10). The `checksec` command results showed that this program only enabled Stack No-eXecute (NX) protection. As a result, we can use an ROP chain to forge the stack structure, control the program execution, and achieve the desired effect of any arbitrary command.

**Figure 10.** Program protection mechanism query result.

We construct specific data packets based on the above analysis to write the shellcode into the newly allocated memory area and execute it. These actions allow us to gain console access to the device and execute user-inputted commands, such as the `id` command to view the UID and GID, and the `whoami` command to view the current username information, as shown in Figure 11.

**Figure 11.** Successfully obtaining remote command execution permissions.

### 4.3. Physical World Threats

CAN is a vehicle bus standard designed to allow microcontrollers and devices to communicate with each other within a vehicle without a host computer. The power control instructions of vehicles are transmitted through CAN. However, the CAN protocol contains no direct support for secure communications [18]. The ADAS system is connected to both the in-car Ethernet and CAN simultaneously. If there is a BOF vulnerability in the ADAS, hackers can use it to access the vehicle's CAN and increase the attack surface of the vehicle. We perform some actions on the vehicle (such as pressing the accelerator pedal) and capture messages on the CAN bus at the same time. Afterward, we use an ADAS device to replay these CAN messages and to observe whether the vehicle triggers the corresponding action while replaying the messages. Suppose the vehicle takes the corresponding actions, which indicates that the ADAS BOF vulnerability can interfere with the vehicle's operation. This attack extends the threat from the digital world to the physical world.

Figure 12 presents our experimental environments. We connect a PC to the ADAS via an Ethernet interface (shown by the blue line) and use a CAN analyzer to connect the ADAS's CAN interface to our PC. The purpose of the CAN connection is to capture the communication traffic. The ADAS device is connected to the high-speed CAN bus.

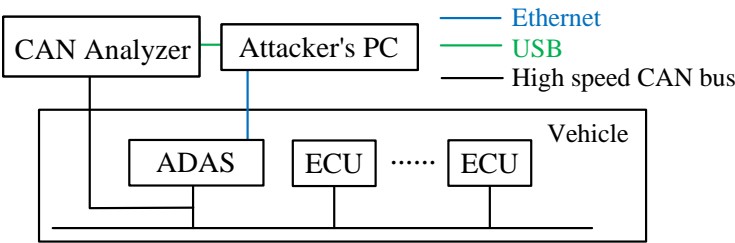

**Figure 12.** Topology diagram of experimental environment.

In this research, by studying and analyzing multiple ADAS devices, we found that ADAS devices are often deeply customized based on the Linux system. Most of them use the Socket CAN module to communicate with the vehicle's CAN bus (the corresponding interface can be seen through the `ifconfig` command, as shown in Figure 13). The application program calls this interface to send and receive CAN data frames.

```
zynq> ifconfig
can0      Link encap:UNSPEC  HWaddr 00-00-00-00-00-00-00-00-00-00-00-00-00-00-00-00
          UP RUNNING NOARP  MTU:16  Metric:1
          RX packets:6771458 errors:0 dropped:0 overruns:0 frame:0
          TX packets:0 errors:6771458 dropped:0 overruns:0 carrier:0
          collisions:0 txqueuelen:1000
          RX bytes:54171664 (51.6 MiB)  TX bytes:512 (512.0 B)
          Interrupt:22

can1      Link encap:UNSPEC  HWaddr 00-00-00-00-00-00-00-00-00-00-00-00-00-00-00-00
          UP RUNNING NOARP  MTU:16  Metric:1
          RX packets:32688216 errors:32687697 dropped:0 overruns:0 frame:0
          TX packets:0 errors:0 dropped:0 overruns:0 carrier:0
          collisions:0 txqueuelen:1000
          RX bytes:261505728 (249.3 MiB)  TX bytes:512 (512.0 B)
          Interrupt:23

eth0      Link encap:Ethernet  HWaddr
          inet addr:192.168.198.198  Bcast:192.168.198.255  Mask:255.255.255.0
          UP BROADCAST RUNNING MULTICAST  MTU:1500  Metric:1
          RX packets:1492 errors:0 dropped:0 overruns:0 frame:0
          TX packets:155 errors:0 dropped:0 overruns:0 carrier:0
          collisions:0 txqueuelen:1000
          RX bytes:87560 (85.5 KiB)  TX bytes:18707 (18.2 KiB)
```

**Figure 13.** ADAS device executing the `ifconfig` command.

We first connect the CAN analyzer to the ADAS's CAN interface and start capturing CAN messages using the corresponding software, as shown in Figure 14. During this process, we perform mechanical operations from the driver's seat, such as stepping on the accelerator pedal. We then export the captured CAN messages and save them as a CSV file during this process.

| Index | System Time | Time Stamp | Channel | Directio | Frame ID | Type | Format | DLC | Data |
|---|---|---|---|---|---|---|---|---|---|
| 00000 | 11:45:04.034 | 0x1B054CC | ch1 | Receive | 0x | Data | Extende | 0x08 | x\| 60 |
| 00001 | 11:45:04.004 | 0x1B05343 | ch1 | Receive | 0x | Data | Extende | 0x08 | x\| 80 |
| 00002 | 11:45:04.034 | 0x1B05487 | ch1 | Receive | 0x | Data | Extende | 0x08 | x\| 50 |
| 00003 | 11:45:04.034 | 0x1B05521 | ch1 | Receive | 0x | Data | Standar | 0x08 | x\| 00 |
| 00004 | 11:45:04.004 | 0x1B0537F | ch1 | Receive | 0x | Data | Extende | 0x08 | x\| 00 |
| 00005 | 11:45:03.974 | 0x1B052C9 | ch1 | Receive | 0x | Data | Extende | 0x08 | x\| 00 |
| 00006 | 11:45:03.974 | 0x1B052C6 | ch1 | Receive | 0x | Data | Extende | 0x08 | x\| 00 |

**Figure 14.** Collecting data from the vehicle CAN.

In Section 4.2, we gained console privileges through ROP and achieved RCE. Additionally, we know that the CAN protocol does not directly support secure communication. In this situation, we exploited the BOF vulnerability on the ADAS to execute a program (as shown in Figure 15) that reads CAN messages from the CSV file and sends them to the vehicle via the SocketCAN module in the ADAS. After running the program to send CAN messages from the CSV file, we observed the vehicle's reaction and confirmed that the wheels turned (as shown in Figure 16).

This result proves the feasibility of obtaining device control through an ADAS BOF vulnerability and sending CAN messages to a vehicle through an ADAS device. Moreover, this means that hackers can disturb a vehicle's operation with the use of an ADAS with BOF vulnerabilities when the network is reachable. This extends the threat from the digital world to the physical world. Our experiment relies on the assumption that the ADAS has network accessibility. If an attacker connects to the in-car Ethernet using either T-box or Bluetooth, it can result in a privacy breach for the users [19]. By employing the method proposed in Section 4.2 to exploit the BOF vulnerability in the ADAS and gain control privileges, the attacker can use the ADAS to remotely control the vehicle by sending CAN commands. This can be achieved without physical contact with the vehicle and on the basis of the analysis of the CAN protocol.

```
C:\Windows\System32\cmd.exe - nc 192.168.198.198 4444

C:\Users\test\Desktop\netcat-win32-1.11\netcat-1.11>nc 192.168.198.198 4444
cd root
./can1_control
this is a can send demo
can_id  = 0x
can_dlc = 8
data[0] = 136
data[1] = 0
data[2] = 0
data[3] = 236
data[4] = 0
data[5] = 0
data[6] = 1
data[7] = 0
```

**Figure 15.** A program to send CAN frames through SocketCAN.

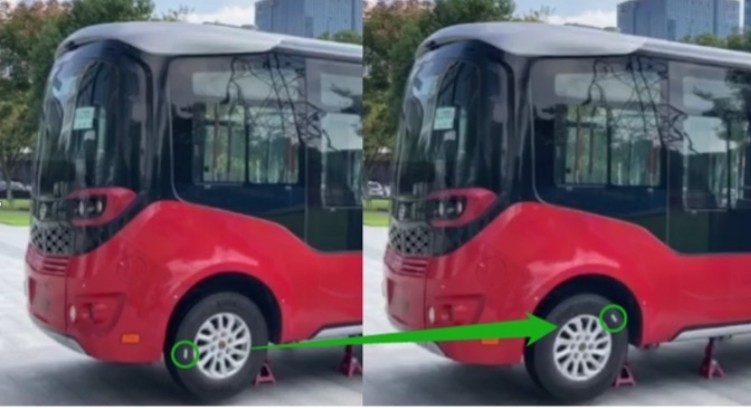

**Figure 16.** Changes in wheel position before and after sending CAN messages.

## 5. Conclusions

Researching ADAS security faces numerous challenges, including a huge code base, varying code quality of different components, and strict confidentiality of related codes and data. To address these challenges, this work proposes a method for mining BOF vulnerabilities in ADAS devices using a communication-traffic-assisted approach. This method involves capturing communication traffic, extracting firmware, analyzing and locating operating systems and applications, identifying risk points based on the communication traffic, tracking data flow, and performing a static analysis. We utilized this method to identify BOF vulnerabilities in commercial ADAS devices and employed ROP to achieve remote command execution. Additionally, we developed a remote vehicle control procedure through a reverse engineering analysis of CAN bus traffic and BOF vulnerabilities and proved its feasibility on actual vehicles.

It should be noted that current research often assumes accessibility to the ADAS's local area network, which may not be feasible in real attack scenarios. Merely relying on BOF vulnerabilities for attacking vehicles is insufficient. It is essential to explore wireless access methods to enhance the effectiveness of attacks. Future efforts will involve exploring more feasible remote attack paths, such as integrating security research on T-Box or in-vehicle entertainment systems, to establish a comprehensive attack chain. This approach aims to enable wireless access to a vehicle's internal network and gain control over the vehicle.

**Author Contributions:** Conceptualization, M.L. In addition, Y.L.; methodology, M.L. In addition, Y.L.; validation, C.C. In addition, J.L.; formal analysis, J.L. In addition, Y.L.; investigation, C.C.; resources, Y.L. In addition, J.L.; data curation, M.L. In addition, J.L.; writing—original draft preparation, M.L., Y.L. In addition, J.L.; writing—review and editing, C.C.; visualization, J.L.; supervision, C.C.; project administration, J.L.; funding acquisition, Y.L. All authors have read and agreed to the published version of the manuscript.

**Funding:** This research was funded by Henan Science and Technology Major Project (No. 221100240100), Shanghai Automotive Industry Science and Technology Development Foundation, SongShan Labtory Pre-Research Project (No. YYJC042022016), Shanghai Science and Technology Innovation Action Plan (No. 21511102500), National Science Foundation of China (No. 62002213) and Shanghai Sailing Program (No. 21YF1413800 and No.20YF1413700).

**Data Availability Statement:** The data that support the findings of this paper are available from the corresponding author upon reasonable request.

**Conflicts of Interest:** The authors declare no conflict of interest.

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
