# Peer review of "Communication-Traffic-Assisted Mining and Exploitation of Buffer Overflow Vulnerabilities in ADASs"

_futureinternet, doi:10.3390/fi15050185_

Round 1
Reviewer 1 Report
This is an interesting and practical article to the emerging popular use of ADAS. It was well prepared and written with sufficient references and appropriate methodology used. The only minor flaw of this article is just showing BOF which is the tip of the iceberg. Suggest the authors to introduce the nature of the various ADAS security problems and their vulnerabilities first and then drill down into the details about BOF.
Basically, this article was well written but there are rooms for improvement in the use of English. Some sentences have duplication of meaning. Better to be concise while rich in contents.
Author Response
Comment A.1:
This is an interesting and practical article to the emerging popular use of ADAS. It was well prepared and written with sufficient references and appropriate methodology used.
Response A.1:
Thank you for your comments.
Comment A.2:
The only minor flaw of this article is just showing BOF which is the tip of the iceberg. Suggest the authors to introduce the nature of the various ADAS security problems and their vulnerabilities first and then drill down into the details about BOF.
Response A.2:
Thank you for the useful comments. we added a paragraph to summarize various security problems.
It has been shown that Advanced Driver Assistance Systems (ADAS) are vulnerable to various attacks that not only compromise their functionality but also pose significant risks to the physical world. For instance, an attacker could tamper with sensor inputs [1], such as radar or lidar, leading to false detections or blocking the system's ability to accurately recognize obstacles [2]. By exploiting vulnerabilities in the ADAS control algorithms, attackers can potentially manipulate steering, braking, or acceleration functions, endangering the lives of occupants and pedestrians [3]. Furthermore, software vulnerabilities enable adversaries to gain unauthorized access to the ADAS network and inject malicious commands or modify crucial system parameters, compromising the overall integrity and safety of the vehicle [4,5].
[1] Nassi, Ben, et al. "Phantom of the adas: Securing advanced driver-assistance systems from split-second phantom attacks." Proceedings of the 2020 ACM SIGSAC conference on computer and communications security, Online, 9-13 November 2020; 293–308
[2] Cao Y, C.Y.; Xiao C, X.C.; Cyr B, C.B. Adversarial sensor attack on lidar-based perception in autonomous driving. Proceedings of the 2019 ACM SIGSAC conference on computer and communications security, London, United Kingdom, 11-15 November 2019; 2267-2281
[3] Ziwen Wan, Z.W.; Junjie Shen, J.S. Too afraid to drive: Systematic discovery of semantic DoS vulnerability in autonomous driving planning under physical-world attacks. arXiv e-prints 2022, submitted.
[4] Tencent Keen Lab: Mercedes-Benz Automotive Information Security Research Overview Report. Available online: https://keenlab.tencent.com/zh/2021/05/12/
Tencent-Security-Keen-Lab-Experimental-Security-Assessment-on-Mercedes-Benz-Cars/. (accessed on 31 Mar. 2023).
[5] Weinmann, Ralf-Philipp, and Benedikt Schmotzle. "TBONE–A zero-click exploit for Tesla MCUs." White paper, ComSecuris (2020).
Reviewer 2 Report
The paper proposes a communication traffic assisted ADAS BOF vulnerability mining and exploitation method to address the complexities of BOF vulnerability mining in ADAS devices. The processes of the suggested method include firmware extraction, analysis of the firmware and system, identification of potential vulnerabilities based on network traffic, validation, and exploitation. This method has been utilised to identify BOF vulnerabilities in commercial ADAS devices and employed ROP to achieve remote command execution. A remote vehicle control procedure was developed through reverse engineering analysis of CAN bus traffic and BOF vulnerabilities and proved its feasibility on actual vehicles.
The research problem is defined clearly, and the paper's contributions are well-described and demonstrated. The steps of the suggested method are shown in detail and in clear visual form. Practical results and physical world threats were demonstrated and discussed systematically.
The paper needs to add a reference citation for this sentence in page 4 ‘In 1988, Robert Morris’s Morris worm virus exploited a BOF vulnerability, causing more than 6,000 network servers worldwide to cash’. Future work is required in the conclusion section.
Author Response
Comment B.1:
The paper proposes a communication traffic assisted ADAS BOF vulnerability mining and exploitation method to address the complexities of BOF vulnerability mining in ADAS devices. The processes of the suggested method include firmware extraction, analysis of the firmware and system, identification of potential vulnerabilities based on network traffic, validation, and exploitation. This method has been utilised to identify BOF vulnerabilities in commercial ADAS devices and employed ROP to achieve remote command execution. A remote vehicle control procedure was developed through reverse engineering analysis of CAN bus traffic and BOF vulnerabilities and proved its feasibility on actual vehicles.
Response B.1:
Thank you for your comments.
Comment B.2:
The research problem is defined clearly, and the paper's contributions are well-described and demonstrated. The steps of the suggested method are shown in detail and in clear visual form. Practical results and physical world threats were demonstrated and discussed systematically.
Response B.2:
Thank you for your comments.
Comment B.3:
The paper needs to add a reference citation for this sentence in page 4 ‘In 1988, Robert Morris’s Morris worm virus exploited a BOF vulnerability, causing more than 6,000 network servers worldwide to cash’.
Response B.3:
Thank you for the advice. We have added the appropriate reference [16] to this sentence in page 4.
[16] Cowan C, Pu C, Maier D, et al. Stackguard: automatic adaptive detection and prevention of buffer-overflow attacks. USENIX security symposium 1998, 98, 63-78.
Comment B.4:
Future work is required in the conclusion section.
Response B.4:
Thank you for the advice. we have expanded the conclusion to include a more comprehensive discussion on the future prospects of our research.
It should be noted that current research often assumes accessibility to the ADAS's local area network, which may not be feasible in real attack scenarios. Merely relying on BOF vulnerabilities for attacking vehicles is insufficient. It is essential to explore wireless access methods to enhance the effectiveness of attacks. Future efforts will involve exploring more feasible remote attack paths, such as integrating security research on T-Box or in-vehicle entertainment systems, to establish a comprehensive attack chain. This approach aims to enable wireless access to a vehicle's internal network and gain control over the vehicle.